# The Long and Short of It: The Emerging Roles of Non-Coding RNA in Small Extracellular Vesicles

**DOI:** 10.3390/cancers12061445

**Published:** 2020-06-02

**Authors:** Agata Abramowicz, Michael D Story

**Affiliations:** 1Maria Sklodowska-Curie National Research Institute of Oncology, Gliwice Branch, 44-102 Gliwice, Poland; agata.abramowicz@io.gliwice.pl; 2Department of Radiation Oncology, University of Texas Southwestern Medical Center, Dallas, TX 75390, USA

**Keywords:** small extracellular vesicles, exosomes, non-coding RNA, lncRNA, Y RNA, piRNA, tRNA fragments, rRNA, circRNA, miRNA

## Abstract

Small extracellular vesicles (EVs) play a significant role in intercellular communication through their non-coding RNA (ncRNA) cargo. While the initial examination of EV cargo identified both mRNA and miRNA, later studies revealed a wealth of other types of EV-related non-randomly packed ncRNAs, including tRNA and tRNA fragments, Y RNA, piRNA, rRNA, and lncRNA. A number of potential roles for these ncRNA species were suggested, with strong evidence provided in some cases, whereas the role for other ncRNA is more speculative. For example, long non-coding RNA might be used as a potential diagnostic tool but might also mediate resistance to certain cancer-specific chemotherapy agents. piRNAs, on the other hand, have a significant role in genome integrity, however, no role has yet been defined for the piRNAs found in EVs. While our knowledgebase for the function of ncRNA-containing EVs is still modest, the potential role that these EV-ensconced ncRNA might play is promising. This review summarizes the ncRNA content of EVs and describes the function where known, or the potential utility of EVs that harbor specific types of ncRNA.

## 1. Introduction

Small extracellular vesicles (EVs) have been extensively studied for about two decades because of the interest in their possible biological roles. Initially, EVs were simply perceived as recycling bins [1], until it was suggested that they could be involved in intercellular communication. It was interesting not only from the point of view of basic studies but also translational research, because there was more and more evidence that they were not immune neutral [2]. Indeed, over the years it became clear that extracellular vesicles have much greater potential than initially assumed. For example, small EVs are known to modulate immune response as either activators and suppressors [3,4] as well as participate in the development of cancer [5,6] or neurodegenerative diseases [7,8]. EVs are mentioned more often in the context of diseases, however, the process of release into the extracellular space is completely natural and is a part of the physiological processes within all body tissues [9]. This phenomenon makes biofluids a very complex mixture of different types of extracellular vesicles (exosomes, microvesicles, apoptotic bodies), that are themselves heterogeneous [9,10].

To date, a major limitation in resolving the biological role of extracellular vesicles was the lack of a method of isolation that would allow for the effective separation of individual groups of EVs based on their origin, instead of their estimated size. Indeed, according to the recent recommendations of the International Society for Extracellular Vesicles [11], a more general nomenclature should be used when describing the results of current EV-based studies, because it is currently not possible to clearly state the origin of EVs in a complex mixture.

Exosomes are small vesicles with a diameter of around 100 nm that is released into the extracellular space by the fusion of multivesicular bodies with the plasma membrane [12]. The tissue of origin is responsible for the specific molecular composition of exosomes. Within the phospholipid bilayer of exosomes, one can find a cargo of proteins, metabolites, and nucleic acids. Some common components of exosomes include membrane proteins like TSG101 and Alix, which are associated with exosome biogenesis, while other components reflect the specific parental cell composition—reticulocyte-derived transferrin receptor; T-cell-derived CD3; and antigen presenting cell-derived MHC class II and I molecules [12,13]. However, exosomes are not a passive reflection of the composition of parental cells. There are well characterized active sorting mechanisms dependent upon or independent of the Endosomal Sorting Complex Request for Transport (ESCRT) machinery [12,14,15]. Other extracellular vesicles, called microvesicles are shed directly from the plasma membrane and can vary in size significantly (from 50 nm up to 1 µm in diameter) [16].

The non-random composition of EVs and the potential biological consequences of this cargo has made decoding their cargo a research priority. While the initial attention for EV cargo was primarily focused on proteins, the discovery of biologically active RNA particles in small EVs by Valadi et al. [17] in 2007 expanded the potential role of EVs in biology. To date, the most extensively examined RNA cargo is miRNA, however, other types of non-coding and coding RNA has been identified. The ability to transmit genetic information or molecules capable of directly modulating the expression of this information was reserved for viruses and bacteria. However, the discovery of biologically active molecules like mRNA and miRNA in extracellular vesicles released in physiological conditions by almost all mammalian cells, expands the complexity of organismal communication systems at the intra- and extracellular/microenvironmental levels, which are far from being fully understood. This complexity has been heightened given recent reports revealing significant representation of other non-coding species of RNA present in extracellular vesicles like t-RNA, rRNA, lncRNA, and piRNA [18]. This review is meant as a summary of the latest studies concerning the role(s) of non-coding RNA content in small EVs with the special emphasis on their diversity and potential for further research in the field of extracellular vesicles.

## 2. Extracellular Vesicles as Carriers of Non-Coding RNA

The most detailed knowledge about the presence of various species of RNA in small EVs comes from next generation sequencing studies. The available data are rather consistent in that EVs are enriched in different types of non-coding RNA (ncRNA) species. The most frequently reported molecules belong to the rRNA, tRNA, and miRNA families, however, Y RNA, piRNA, snRNA, snoRNA, and lncRNA are also quite common [18,19,20,21,22,23,24,25,26,27,28,29]. Some studies have identified the presence of tRNA-derived RNA fragments (tRFs) [29], small Cajal body-specific RNA (scaRNA) [18,26], long intergenic non-coding RNA (lincRNA) [24,25], small cytoplasmic RNA (scRNA) [27], or other miscellaneous RNA (miscRNA) [19,25], and there is no reason to assume that the total list of small vesicle RNA is currently complete. Interestingly, there is no consensus of the percentage content of particular classes of RNA in small EVs.

Some studies suggest that the most represented RNAs are microRNA molecules (about 50% to 70%) [20,21,24,28,29], however, Wei et al. [27] following Raabe et al. [30] suggested that most commonly used protocols for small RNA library constructions prefer miRNA molecules and it might significantly affect the results. In fact Wei et al. [27] suggested that miRNA accounts for less than 10% of the total RNA, with about 90% of EV-related ncRNA being rRNA molecules. A similar observation was made by Jenjaroenpun et al. [22]. On the other hand Baglio et al. [19] reported that the highest enriched fraction of ncRNA is tRNA or miscRNA, depending on the cell line examined. Y RNA was identified as the most enriched ncRNA in plasma-derived EVs from lung cancer patients [23]. In addition, protein coding RNA was overrepresented in small vesicles isolated from archival human serum [25]. A major reason for these differences in comparing the cargo of EVs is likely the plurality of the strategies for EV isolation [31]. However, most reports described here were based on small EVs collected by precipitation [18,20,21,23,24,25,26,28,29]. This indicates that in studies of extracellular RNA, the RNA extraction or library preparation methodologies, the sequencing parameters themselves or bioinformatics processing could also affect the final conclusions. However, comparative studies presented by Schageman et al. [26] and Li et al. [18] suggest that representation of particular groups of RNA in small EVs might be strictly dependent on the source of EVs. For example, the profiles of RNA content in small EVs derived from HeLa cells and human serum differ [26], as it does when comparing serum- and urine-derived vesicles processed by the same methodology [18]. Without doubt, the actual contribution of individual RNA fractions in EVs requires further study, considering the potential impact of various factors on the final experimental results.

Besides preparation methodologies, one of the more serious challenges in studies of RNAs in EVs is the low content of isolated nucleic acids. Studies of Eldh et al. [32] and Prendergast et al. [25] described a significant dependence of the total RNA yield on the method of EV and RNA isolation. The comparison of seven methods of total RNA extraction from small EVs isolated from cell culture supernatants by ultracentrifugation yields, ranged from 13.0 (±7) to 107.7 (±25.7) ng per million donor cells (as determined by an Agilent 2100 Bioanalyzer), with the highest efficiency for the miRCURY (Exiqon) isolation kit [32]. When using 500 µl of serum, the yield of RNA ranged from about 129 to over 1100 ng in the case of the ExoQuick-mediated enrichment of EVs and from about 145 to about 195 ng of RNA when ultracentrifugation was used to isolate EVs (yield determined by Nanodrop) [25]. The highest efficiency in this study was generated through use of the ExoQuick system used in conjunction with traditional RNA precipitation, via phase separation. In other studies ultra-centrifugation of 250 µl of human plasma allowed for isolation of 0.2–1.1 × 10^8^ of small EVs that resulted in the isolation of 10–15 ng of total RNA [21]. Furthermore Li et al. [18] reported that EVs isolated by precipitation from 4 mL of serum contained 2–10 ng of total RNA, while 10 mL of urine resulted in about 2–4 ng. Besides quantitative and stoichiometric analysis of the miRNA content of small EVs, even the most abundant species of miRNA are present in less than one copy per vesicle, and within the EV population, only some EVs contain particular sequences of miRNA in very low concentration [33].

A separate issue is a reliable quantitative assessment of the concentrations of small RNA molecules. A comparison of quantities isolated by few more popular techniques of RNA quantification obtained for miRNA isolated from serum samples revealed significant differences among these methods [34]. In general, spectrophotometers overestimated miRNA content by detecting contaminants. The Agilent 2100 Bioanalyzer Pico Chip and Small Chip were able to properly specify the profile of RNA, but failed to quantify, while the Qubit 2.0 Fluorometer provided the most accurate quantification of RNA content but failed to profile the RNA. In fact, according to the ISEV position paper [35], there is no one dedicated method for the quantification of the RNA cargo of EVs, since all available methods have some limitations (for example, too high of a lower detection threshold, sensitivity to DNA contamination or no total RNA quantification). However, in general the Nanodrop device is not recommended for the EV-RNA quantification and the pre-treatment of samples with DNase is strongly suggested for accurate RNA quantitation. Currently, technical limitations is one of the biggest challenges in EVs-related studies because of the low content of specific cargo components, however, technological improvements continue.

## 3. miRNA

MicroRNAs (miRNA) are small non-coding RNAs composed of 21–25 nucleotides that are responsible for the regulation of gene expression. Maturation of the molecule includes several stages, like transcription into hairpin-structured pri-miRNa and further processing into pre-miRNA hairpins consisting of 70 nucleotides [36,37]. In the cytoplasm, the pre-miRNA is cleaved by Dicer RNase into the specific length and then one strand of mature miRNA binds to the Argonaute complex and the other one is targeted for degradation. The canonical role of miRNA is regulation of gene expression that might be performed at the post-transcriptional level by repression of translation or induction of mRNA degradation, however, a regulatory activity for nuclear miRNA was recently recognized. For more details see reviews [38,39]. The presence of non-randomly packed miRNA in EVs was initially described in EVs secreted from MC/9 mouse liver mast cells [17], catapulting EV research within the context of cellular response to stress or biomarker research. The influence of ionizing radiation, a cellular stressor, on the composition of EV miRNA was described in the studies of EVs isolated after irradiation of whole blood samples with 2Gy [40]. While there was no difference in the total number of miRNAs identified in irradiated vs. non-irradiated samples, irradiation significantly increased the level of three miRNAs, miR-204-5p, miR-92a-3p, and miR-31-5p, known regulators of genes involved in apoptosis, immune response, and cell proliferation. Irradiation of glioma cells with 3 and 12 Gy significantly affected four of 516 miRNAs found in EVs released in vitro. These miRNA included the downregulation of tumor suppressive miRNAs miR-516 and miR-365 and upregulation of miR-5588 and oncogenic miR-889 [41]. In addition, the direct involvement of EVs in the radiation-induced bystander effect (RIBE), through the delivery of miR-21 and miR-7-5p was suggested for the human normal embryonic lung fibroblast cell line (MRC-5) [42] and the human bronchial epithelial cell line (BEP2D) [43]. Further analysis of miRNA profiling of EVs released under hypoxic conditions by the human prostate cancer cell line LNCaP, revealed a significant influence of oxygen deficiency on extracellular vesicles composition [44]. Among 292 miRNA identified, there were 11 miRNAs (miR-517a, miR-204, miR-885, miR-143, miR-335, miR-127, miR-542, miR-433, miR-451, miR-92a, and miR-181a) that were significantly increased and 9 miRNAs (miR-521, miR-27a, miR-324, miR-579, miR-502, miR-222, miR-135b, miR-146a, and miR-491) that were significantly decreased as a result of hypoxia. Two of these miRNAs (miR-885 and miR521) were also similarly expressed in EVs isolated from the serum of prostate cancer patients [44]. miR-193a-3p, miR-210-3p, and miR-5100 were found in small EVs released by bone marrow-derived mesenchymal stem cells (BMSCs) under hypoxic conditions; these are recognized to be supporters of invasiveness in recipient cancer cells by activation of the STAT3 signaling [45]. let-7a miRNA was identified as the molecule responsible for hypoxic-EV-mediated enhancement of mitochondrial oxidative phosphorylation in macrophages [46], while miR-210-3p released in EVs from hepatocellular carcinoma cells promotes angiogenesis in endothelial cells by the inhibition of SMAD4 and STAT6 [47]. The active role of EV-delivered miRNA was also observed in the development of viral infection [48], where miR-146a selectively packaged into EVs by infected cells was able to suppress type I interferon response in recipient cells, thus, facilitating the transmission of EV71 virus.

As important components of the cargo of EVs, miRNA are also considered as promising targets for translational research. Recently, several reports described the use of EV-related miRNA as biomarkers for different diseases. For example, based on the comparison of the profiles of EV-associated miRNA isolated from the serum of patients with recognized relapsing–remitting multiple sclerosis (RRMS) treated with the immunomodulatory drug INF-β, or not, a panel of 16 miRNas were identified with potential prognostic value for therapeutic response in patients with multiple sclerosis [49]. Similarly, a prognostic potential for circulating encapsulated microRNA was also suggested for cancer research. The analysis of 156 plasma samples from multiple melanoma patients led to the identification of let-7b and miR-18a as significantly associated with both progression-free survival and overall survival [24], whereas miR-30d, miR-140, and miR-29b were recognized as significantly associated with survival of hepatocellular carcinoma patients [50]. Furthermore, examination of miRNA content in urinary EVs revealed promising candidates for prostate cancer diagnosis. It was shown in two independent cohorts of patients that miR-196a-5p and miR-501-3p were significantly downregulated in EVs isolated from the urine of prostate cancer patients. These results required further research with a larger number of patients [51].

## 4. circRNA

Circular RNA (circRNA) is a single strand RNA formed as the closed continuous loop with covalently linked ends [52]. Based on the origin, it can be classified as exonic circRNA, circular intronic RNA (ciRNA), exon-intron circRNA (EIciRNA), or intergenic circRNA [53,54]. The most abundant class of circRNAs in eukaryotic cells are cytoplasmic exonic particles with linked 3′ and 5′ ends [55]. In fact, though most of the circRNAs include sequences of active genes, they generally do not have coding properties, with some individual exceptions that are capable of being translated to short peptides [56]. In fact, these specific sequences allow circRNAs to target their complementary miRNAs to deactivate them (microRNA sponging) [57]. For example, the ciRS-7 has nearly 70 miRNA target sites for an efficient suppression of mir-7 activity and thus the promotion of the expression of mir-7 targets [57]. Moreover, circRNAs can also perform regulatory functions through direct interaction with proteins (protein sponging) [58]. A well-known example is circ-Foxo3, which forms a complex with crucial cell cycle regulators p21 and CDK2, leading to the inactivation of CDK2 and cell cycle arrest [58]. Some specific features of circRNAs like exceptional stability through resistance to exonucleases and expression in a tissue-dependent manner (not correlated with the level of linear mRNA) [52,59] make them promising subjects of study in translational research, especially in the biomarker field (see the comprehensive reviews by Zhang et al. [60] and Lei et al. [54]). Moreover, the identification of the presence of stable circRNAs in extracellular vesicles also sheds new light on the functional study of both subjects [61]. It was shown that the circular forms were not only more abundant in EVs than their linear mRNA isoforms but were also generally significantly enriched in vesicles, compared to parental cells [61,62]. Current studies are frequently focused on the role of circRNAs in EV-mediated disease development, as well as on the potential usage of encapsulated circRNAs as biomarkers in EV-based liquid biopsies. Indeed, recent studies reveal the importance of EV-related circRNA for cancer progression and resistance to treatment. Zhang et al. [63] showed that circ-DB delivered by adipocyte-related EVs can support the growth of hepatocellular carcinoma and reduce DNA damage by suppressing mir-34a and activating the USP7/Cyclin A2 signaling pathway. It was also shown in in vivo studies that the circNRIP1 when significantly elevated in EVs from the plasma of gastric cancer patients can promote EMT and metastasis [64]. Furthermore, Wang et al. [65] determined that EVs from chemoresistant-cells can enhance the resistance of sensitive cells by EV-mediated transfer of ciRS-122, which decreases the level of miR-122 and therefore upregulates PKM2 and glycolysis. In other studies, Zhao et al. [66] analyzed the circRNA content of EVs derived from radioresistant glioblastoma cells and proposed circATP8B4 as a potential supporter of the development of resistance to radiation in glioblastoma recipient cells. In the field of biomarkers, specific profiles of circRNA in EVs isolated from the serum of patients with papillary thyroid carcinoma [67] and endometrial cancer were developed [68]. Both pioneering studies provided promising results that showed the high potential of EV-related circRNAs as potential biomarkers in liquid biopsies. In fact, circRNA is a relatively new subject of study, especially those associated with EVs, and might be key to understanding any number of biological phenomena.

## 5. tRNA

Transfer ribonucleic acid (tRNA) are small molecules consisting of 76–93 nucleotides that form a characteristic cloverleaf structure terminated by the CCA trinucleotide at the 3′-end with an acceptor stem, a dihydrouridine (D) stem-loop, an anticodon stem-loop, and a TψC stem-loop [69]. tRNA is one of the most abundant RNA, as there are about 300 cytoplasmic tRNA molecules with different nucleic acid sequences [69]. The main and the best understood function of tRNAs is the participation in translation by carrying amino acids to the ribosome during protein synthesis. However, fragmentation of tRNA molecules is the source of two t-RNA-related families of ncRNA with regulatory function: tiRNAs (interfering tRNAs) and tRFs (tRNA-derived fragments).

tiRNA molecules are created by the cleavage of the tRNA molecule in the anticodon loop, performed by angiogenin, during stress conditions [70]. The 3′ and 5′ halves consist of 30–50 nucleotides that are involved in the stress-dependent formation of stress granules [71,72], inhibition of transcription [70], and regulation of apoptosis, by capturing cytochrome c and preventing the initiation of cell death [73]. Conversely, tRFs are shorter fragments of tRNA than tiRNA, consisting of 16–28 nucleotides that can be classified on the basis of their sites of origin into three main types: (i) tRF-5, derived by a cleavage in the D or anticodon loop, (ii) tRF-3, formed from the 3′-end of a mature tRNA, usually by cleavage in the T loop, (iii) and tRF-1, derived from the pre-tRNA after RNase Z cleavage at the poly-U residues end. Some specific tRF fragments were recognized as molecules involved in the inhibition of protein translation [74], progression of retroviral infection as primers for reverse transcription [75], or in the regulation of cell proliferation and DNA damage response by inhibition of the RPA protein [76]. Some tRNA fragments were suggested as potential markers of trastuzumab resistance in breast cancer patients [77] and indicators of progression-free survival, and as a prognostic marker for prostate cancer patients [78].

All three classes of tRNA were identified in small EVs. Shurtleff et al. [79] performed several experiments using RNase and DNase treatment in the presence or absence of detergent and showed that tRNAs were the most abundant species of RNA in small EVs released from HEK293T cells, although the mature full length molecules dominated over tRNA fragments. Furthermore, an RNA-binding protein called Y-box binding protein 1 (YBX1) and posttranscriptional modification at the nucleotide usually occupied by dihydrouridine, were proposed as a part of the mechanism of selective tRNA packaging to EVs [79]. Studies of the RNA content of small EVs derived from mesenchymal stem cells also revealed that vesicles were highly enriched in tRNA particles (over 50% of total small RNA in adipose-derived exosomes and 23–35% in bone marrow-derived EVs) [19]. However, the content of specific subclasses of tRNA molecules was dependent on the parental cell type. EVs derived from adipose and more differentiated bone marrow stem cells carried mostly tiRNA (5′ halves) and no full length mature tRNA, whereas less differentiated bone marrow stem cells contained both mature molecules and fragments. Significant enrichment of tRFs (especially tRFs-5) fragments as part of a comparison with cellular RNA content was observed in small EVs released from activated T cells [80]. Additionally, it was shown that some of those molecules like tRFs-5 derived from the tRNAs Leu-TAA and Leu-TAG specifically loaded into EVs, in response to T cell activation. The diagnostic potential of tRNA fragments was also investigated in patients with osteoporosis [29]. A model composed of 3 different tRF molecules was proposed for the diagnosis of osteoporosis with an AUC of 0.815.

## 6. Y RNA

Y RNAs are small non-coding RNAs represented in humans by four types of particles—hY1, hY3, and hY4 that are found predominantly in the cytoplasm, although also in the nucleus [81]; and hY5 whose localization is predominantly nuclear [82]. The single molecule consists of 83–112 nucleotides that form a characteristic structure with a polyuridine tail (for binding La protein), a lower stem domain (to bind the Ro60 protein), an upper stem domain (involved in initiation of DNA replication), and a loop domain (which is the most heterogeneous, modulates chromatin association, and can be a binding site for some interacting proteins) [81]. The function of Y RNA is not well understood, although several hypotheses were proposed. As Ro60 binds misfolded ncRNA, it is suspected that Y RNA might be involved in RNA quality control as a structure regulating the access of RNA to Ro60 protein [83,84] or recognizing the misfolded ncRNA, and recruiting appropriate enzymes [84]. Another presumed function for Y RNA is participation in the initiation of chromosomal DNA replication (especially Y1 and Y3 RNA) [85] and support for cell proliferation [86]. Unfortunately, not much is known about the function of Y RNA-derived fragments, other than the fact that they are produced in apoptotic and non-stressed cells in both cancer and non-cancerous cells [87], by cleavage within the internal loop domain [88]. Both full length Y RNAs and their fragments are considered as potential biomarkers. All types of full length Y RNAs (especially hY1 and hY3 RNA) were seen to be significantly overexpressed in solid tumors, when compared to the corresponding normal tissues [86] and specific Y RNA fragments isolated from serum, have the potential to be useful in breast cancer diagnosis [89].

Deep sequencing of EV RNA content confirms the presence of Y RNA in small vesicles with the percentage of Y RNA particles being relatively high when compared to all non-coding RNAs. In small EVs isolated from the serum of lung adenocarcinoma and lung squamous cell carcinoma patients, over 50% of sequence reads were annotated to Y RNA [23]. Y RNAs and especially their fragments were also highly abundant in EVs from patient-derived glioma stem-like cultures [27]. Y RNA fragments were significantly enriched in extracellular RNA (both in small EVs and ribonucleoproteins), as compared to cellular RNA [27]. Y RNAs were also one of the most commonly identified sequences found in EVs from in vitro culture of laryngeal squamous cell carcinoma (AMC-HN-8 cells) [20].

Interestingly, Y RNA content in EVs from human semen suggests that each type of Y RNA is loaded selectively [90]. The majority of full length and fragment Y RNA was hY4. For hY1, hY3, and hY5 RNA, almost all smaller fragments were derived from the 5′ end of Y RNA, with the exception of hY4, where one-fifth of the smaller fragments were mapped to the 3′ end. Moreover, the ratios of larger to smaller fragmented Y RNA read counts for all classes of Y RNA were consistent in EVs from six donors [90]. The significant overrepresentation of hY4 RNA-derived fragments was also observed in vitro in EVs from cardiosphere-derived cells (CDCs) and normal human dermal fibroblasts (NHDFs) [91]. The most plentiful fragment in CDC-related vesicles was the hY4 RNA fragment from the 5′ end, marked as EV-YF1, and it was almost 16-fold more abundant when compared with NHDFs EVs. It was determined that these particles, delivered via EVs, were able to alter the IL10 gene expression and to enhance the IL-10 protein secretion in targeted macrophages. Furthermore, it was shown that the secretion of IL-10 by bone marrow-derived macrophages stimulated by EV-YF1 fragments had a cytoprotective effect on oxidatively stressed cardiomyocytes, both in vitro and in vivo [91]. Further studies have shown that both CDC-derived EVs and synthetic EV-YF1 treatment was able to attenuate cardiac hypertrophy and renal injury induced by angiotensin II infusion, by altering the anti-inflammatory cytokine IL-10 expression in the heart, spleen, and kidney [92]. Unfortunately, the molecular mechanism of action for Y RNA fragments remains unclear.

## 7. piRNA

Piwi-interacting RNA (piRNA) is an abundant class of small non-coding RNAs with little conservation of sequence between organisms [93]. What is known comes mainly from research on *D. melanogaster, C. elegans*, and *Mus musculus*. piRNA are single-stranded, 21–30 nucleotide molecules transcribed from characteristic piRNA clusters located in euchromatic (mice) or heterochromatic regions (*Drosophila*), which in this case were not transcriptionally silent [93,94,95]. The main function of piRNA is repression of transposons at the transcriptional and posttranscriptional level, to maintain genome integrity [96]. This appears to be a defense mechanism especially crucial for the protection of genetic information in germ cells, where piRNA are highly enriched and predominantly found [93,94]. piRNAs interact with a specific class of Argonaute proteins called PIWI proteins [97]. Depending on the species, PIWI families consist of two to four members that determine whether piRNA is localized to the nucleus or to the cytoplasm [94,97]. The function of piRNA/PIWI complexes seems not to be limited to transposon silencing, as it was shown that PIWI is also required for the epigenetic activation of the sub-telomeric region (3R-TAS) in *Drosophila* [98], and can suppress the expression of genes by binding to their genomic regions; the human melatonin receptor 1A gene (MTNR1A) by piRNA_015520, as an example [99]. Although piRNA are mainly linked with germinal tissues, sets of piRNA were also identified in somatic cells of human brain [99] and mouse hippocampus [100] and are commonly present in human plasma [101]. In addition, both piRNA and PIWI proteins are thought to be involved in cancer development with some of their individuals pathologically expressed in cancers like breast [102] or gastric cancer [103], and are under consideration as biomarkers (reviewed in details by Han et al. [104] and Ng et al. [105]). Piwi-interacting RNA is also commonly identified in extracellular vesicles of various origin like in vitro culture of AMC-HN-8 [72] and HeLa [26] cell lines, human semen [90], or human plasma [21,23,106]. Usually, piRNA content in EVs is approximately 1–4% of all identified sequences. However, Yuan et al. [28] reported piRNA was almost as abundant as miRNA (40% of mapped reads), in EVs isolated from the plasma of 192 individuals (healthy donors and cancer patients). Furthermore, they were able to identify 118 different piRNA particles. Unfortunately, the role of piRNA in extracellular vesicles is unknown. Yang et al. [106] were able to distinguish healthy volunteer- derived samples from patients with heart failure, based on their EV-related piRNA profiles. While the small sample size decreased the significance of this finding, the results were encouraging. Evidence for a direct influence of EV-delivered piRNA on recipient cells was shown by De Luca et al. [107], where they showed that EVs released by bone marrow mesenchymal stem cells (BM-MSC) were able to affect the gene expression profile of umbilical cord blood CD34+ stem cells (UCB-CD34+), making them less susceptible to apoptosis and less differentiated. Based on the results of sequencing of the small RNA present in these EVs, they identified both miRNA and piRNA that are likely to be involved in the regulation of programmed cell death and cellular differentiation processes.

## 8. rRNA

Ribosomal nucleic acid (rRNA) is a major component of the ribosomes–molecular machines responsible for protein synthesis (translation). Human ribosomes have a molecular weight of 4.3 MDa and consist of two subunits—large (60S), which includes three types of rRNA particles; (i) 28S (5064 nt), (ii) 5S (120 nt), and (iii) 5.8S (157 nt) rRNA; and small (40S), which includes a single 18S (1869 nt) rRNA molecule [108,109]. Ribosomal RNA is transcribed from tandemly arrayed units and the sequences encoding 28S, 5.8S, and 18S rRNAs are located in the characteristic nucleolus organizer regions (NORs) [110]. Generally, ribosomes are associated with the endoplasmic reticulum, where they are responsible for the translation of secretory or transmembrane proteins, or they can be free in the cytoplasm, providing synthesis of cytosolic proteins [111]. The high throughput study of Jenjaroenpun et al. [22] on human breast cancer cell lines showed that over 80% of RNA-seq reads from the RNA of small EVs were mapped to rRNA. Similarly, in EVs derived from HeLa cells, about 40% of reads were related to ribosomal rRNA. rRNA was also detected in small EVs derived from body fluids like serum and urine, but with different abundance. In urine, rRNA represented at least half of all identifications, while in serum it was no more than 20% [18]. To date, small EV-delivered rRNA has not shown any biomarker potential and no specific function like vesicle cargo, was identified. On the other hand, there are several reports where no evidence of rRNA in EVs was seen. This included studies of EV derived from human saliva, plasma, and breast milk [112], as well as from in vitro cell culture [17,21,113,114], where no characteristic 18 or 28S rRNA peaks were detectable by Agilent Bioanalyzer (chip-based capillary electrophoresis) profiling. This position was supported by the International Society for Extracellular Vesicles during the workshops in 2012 [115] and 2015 [71]. Indeed, in studies of EV cargo in prion-infected neuronal cells, Bellingham et al. [116] did not find 18 or 28S rRNA peaks when using a Bioanalyzer, however, next generation sequencing did identify EV-5S rRNA. This finding suggests the possibility that, as in case of other species of RNA, specific small rRNA molecules could intentionally be loaded into small EVs like exosomes.

## 9. lncRNA

Long non-coding RNAs (lncRNA) are traditionally defined as transcripts longer than 200 nucleotides with no protein-coding function. However, at least one exception was noted [117]. This broad classification made this group of molecules very diverse [118]. Due to their low expression levels, lncRNAs were considered as “transcriptional noise” with no expectation that these RNA could be key players in the regulation of transcription. While this is not the case today [118,119], lncRNA are still relatively poorly understood, compared to other non-protein-coding transcripts, and their high diversity make them very hard to classify and annotate. So far, the most common classification is based upon genomic location rather than function or other properties [118,120,121]. For example, according to the categorization proposed by Kung et al. [120], five classes of lncRNA can be distinguished—(i) stand-alone RNAs (lincRNAs) with sequences not overlapping with protein-coding genes, (ii) antisense transcripts consisting of RNAs with different degrees of complementarity to the sense transcripts like protein-coding mRNA, (iii) pseudogenes containing lncRNA transcribed from sequences of genes that lost their coding functionality after mutation, (iv) long intronic ncRNAs consisting of RNAs transcribed from the intron of a protein-coding gene, and (v) other lncRNAs containing divergent transcripts, promoter-associated transcripts, and enhancer RNAs.

Although the functions of some lncRNA are quite well-understood, the general knowledge of lncRNA regulation and function is still immature. The known roles for lncRNA include a crucial role in many cellular and developmental processes via chromatin remodeling, transcription regulation, and intercellular trafficking [122]. For example, the nuclear lncRNA GM12371 selectively regulates the expression of genes involved in the development and function of the nervous system and is essential for synaptic transmission [123]. This is achieved by association of GM12371 with transcriptionally active chromatin in the region of the targeted genes. The lncRNA SYISL (SYNPO2 intron sense-overlapping lncRNA), which is highly expressed in skeletal muscle, was identified by Jin et al. [124] as a factor promoting myoblast proliferation and inhibiting myogenic differentiation. SYISL inhibits the transcription of targeted genes by binding to their promoter and recruiting polycomb repressive complex 2 (PRC2) with histone methyltransferase activity. HCCL5 is a cytoplasmin lncRNA frequently overexpressed in human hepatocellular carcinoma cells. It is responsible for promoting cell growth, migration, and metastasis, as well as inhibition of apoptosis, by increasing the expression of the transcriptional factors involved in the epithelial–mesenchymal transition [125]. The broad regulatory capabilities of lncRNA extend to lncRNA in EVs, which after delivery, are able to influence the recipient cell fate.

Based on what is known about EV-derived lncRNA, lncRNAs, like EV-derived miRNAs, might be a key component of the EV cargo, which act as intercellular mediators [126]. The selective packing of lncRNA molecules was shown by Gezer et al. [127] in in vitro studies of HeLa and MCF-7 cell lines under genotoxic stress, where lincRNA-p21 and ncRNA-CCND1-lncRNA, with very low cellular expression levels, were significantly enriched in small EVs and increased in both cells and EVs, in response to bleomycin treatment. Elevated levels of lncRNA-p21 was also observed in small EVs isolated from the urine of prostate cancer patients and was proposed by Işın et al. [128] to be a molecule of high potential to discriminate prostate cancer from benign prostatic hyperplasia. LncRNA GAS5 was identified as a component of the vesicles released from THP-1 cells (spontaneously immortalized monocyte-like cell line), after oxidized low-density lipoprotein (oxLDL) treatment, which is responsible for enhancing apoptosis in recipient vascular endothelial cells [129]. lncRNA RP11-838N2.4 was shown to be significantly upregulated in erlotinib-resistant non-small cell lung cancer cells. Furthermore, when transferred via small EVs to recipient treatment-sensitive cells, erlotinib resistance was induced [130]. Similarly, linc-ROR, which is highly expressed in hepatocellular cancer cells, was also found in significant quantities in HCC-delivered extracellular vesicles and was even more abundant after sorafenib treatment [131]. Depletion of linc-RORs from EVs, significantly increased sensitivity to sorafenib in recipient cells, in comparison to cells incubated with linc-ROR containing EVs, indicating a direct involvement of the delivery of this lncRNA by EVs, with the development of chemoresistance of hepatocellular cancer cells.

## 10. Conclusions

While miRNA is the most extensively studied component of small EV cargo, recent studies show that it is not the only representative of the non-coding transcriptome present in small EVs. The fact that the composition of individual molecules of RNA with very diverse functionalities is dependent on the origin and physiological state of parental cells and that these might be influenced by various environmental factors suggests that there is a distinct need to understand the biological function of the ncRNA cargo of EVs. What is known of the ncRNA species in EVs appears to be particularly promising (Table 1). The presence of miRNA molecules that can be directly involved in the regulation of gene expression, begs the question of the need for EVs as delivery vehicles for ncRNA targets. In fact, the recent discovery of the presence of circular RNA in small EVs and the emerging evidence for their active regulatory role in recipient cells have shed new light on the regulatory capabilities of extracellular vesicles, through the regulation of genes involved in normal developmental processes, as well as cancer progression.

Other molecules with regulatory function include tRNA fragments that are significantly enriched in EVs from some types of cells. Given their role in the context of cellular stress (viral infection, DNA damage response, drug resistance), tRNA fragments appear to be particularly interesting as prognostic or predictive molecules of basic biology or therapeutic response. Similarly, lncRNAs might be important in at least two areas. The first, the EV-mediated resistance to chemotherapeutic agents like lncRNAs delivered by EVs, are able to reduce sensitivity to a number of chemotherapeutic agents like erlotnib and sorafenib. Second, these act as a biomarker for the presence of prostate cancer. Although there is very little knowledge about Y RNA in and of themselves, the level of these molecules in EVs might vary significantly, depending on their origin and physiological state, which makes them a very promising candidate for molecular biomarkers.

Both piRNA and rRNA are families of non-coding RNA that were not necessarily expected to be common components of EVs. While the primary function of piRNAs is genome integrity and were mainly identified in germinal cells, their presence was confirmed in EVs from different sources, although no biological function for piRNA found in EVs has been described. At the same time, rRNA molecules are one of the most controversial components of small vesicle content. Not only is their function unknown, even their presence in EVs is still a subject of discussion.

Last, determining the function of, or consequences of, non-coding small and long RNA being found in extracellular vesicles remains a challenge. Furthermore, it would appear that there might be RNA-binding proteins that might form stable complexes with small RNAs within exosomes, which are responsible for maintaining small RNAs in exosomes as well as shuttling small RNAs between exosomes and recipient cells [132]. With time, we might yet be able to decipher the information contained in EVs for its prognostic or predictive value. This would be especially valuable in the arena of cancer detection and therapy. EV cargo likely provides a clear picture of the human condition. We just do not yet know how to interpret the message contained in EV cargo.

## Figures and Tables

**Table 1 cancers-12-01445-t001:** Selected examples of non-coding RNAs identified in small extracellular vesicles with the recognized function in recipient cell.

RNA	EV Source	Method of EV Purification	Function in Recipient Cells	Ref.
miRNA 210-3p	Hepatocellular carcinoma cells	UC	Promotion of angiogenesis by inhibition of SMAD4 and STAT6 in endothelial cells	[47]
miRNA miR-193a-3p, miR210-3p, miR-5100	Bone marrow-derived mesenchymal stem cells	UC	Promotion of metastasis by activation of STAT3 signaling in lung cancer cells	[45]
miRNA miR-21	Human normal embryonic lung fibroblast cell line MRC-5	Precipitation	Supporting of radiation induced bystander effect (RIBE)	[42]
circRNA ciRS-122	Colorectal cancer cell line SW480	UC	Induction of chemoresistance by decreasing level of miR-122 and stimulation of glycolysis	[65]
circRNA circNRIP1	Gastric cancer cells MKN-45 and BGC-823	Precipitation	Promotion of EMT and metastasis	[64]
Y RNA EV-YF1	Cardiosphere-derived cells	Concentration	Regulation of IL10 gene expression and stimulation of secretion of IL-10 protein by macrophages	[91]
lncRNA GAS5	Spontaneously immortalized monocyte-like cell line THP-1	UC	Regulation of apoptosis in vascular endothelial cells	[129]
lncRNA RP11-838N2.4	Non-small-cell lung carcinoma cell lines HCC827 and HCC4006	Precipitation	Chemoresistance development	[130]
lncRNA linc-ROR	Human liver cancer cell line HepG2	UC	Chemoresistance development	[131]
piRNA	Bone marrow mesenchymal stem cells	UC	Participation in regulation of programmed cell death and cellular differentiation	[107]

UC—Ultracentrifugation.

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
