# Peer review of "The Long and Short of It: The Emerging Roles of Non-Coding RNA in Small Extracellular Vesicles"

_cancers, 2020, doi:10.3390/cancers12061445_

Round 1

Reviewer 1 Report

In this article, the authors discuss in authoritative way the roles of non-coding RNAs in extracellular vesicles (EVs), and understanding their biological roles in health and diseases. Additionally, the prognostic or predictive importance of RNA content in EVs and using these vesicles+ncRNA as therapeutics have been proposed. To provide a balanced review, authors also discuss the limitations in studying EVs and EV associated RNA yield.

The current review is timely covered and provides current knowledge in the field. I enjoyed reading through the content and discussion. I can say that the review is well written, well organized and easy to follow. However, I have minor suggestions for authors to further improve their manuscript.

(1). Page 2: While authors indicate ‘’major limitation in resolving the biologic role of EVs has been the lack of a method of isolation that would allow for the effective separation of individual groups of EVs based upon their origin instead of their estimated size’’. Authors may wish to refer also the obstacles in studying the RNA in EVs, perhaps referring the following article (PMID: 28326170). Perhaps in connection with, one of the more serious challenges in studies of RNAs in EVs is the low content of isolated nucleic acids (total RNA yield) and the method of EV and RNA isolation (Page 3).

(2). The review content itself is worth reading and getting current knowledge on the subject, perhaps it would be worth including a table to give a quick overview to readers?.

- If authors wish to include a table, it can be done by including ncRNA category/class (miRNA, Y-RNA, lncRNA, piRNA etc) in first column, then their source and method of isolation in next column, and finally functions in another column.  

(3). Finally, while EV associated roles of lncRNAs have been described, authors may wish to refer the following article (PMID: 29657282).

On a separate note, in the scope of current subject, the stability of RNA by RNA-binding proteins in EVs (exosomes) may have supportive role to keep the RNA functional. E.g. those reported by Luisa Statello et al 2018. 

Reviewer 2 Report

Abramowicz and Story present an interesting compendium of information about the EV cargos from the literature. The manuscript is well written and clear, giving on top some important cues on possible bias derived from the different methods of EV isolation and RNA sequencing.

Given the growing data indicating the presence of circRNA into EVs, this reviewer would suggest adding a paragraph commenting on that.

Also, it would be beneficial to include a representative schematic figure summarizing the whole content of the manuscript (subcellular origin of small EVs and the different types of cargo loading).
